# Creation of AlSi12 Alloy Coating by Centrifugal Induction Surfacing with the Addition of Low-Melting Metals

**DOI:** 10.3390/ma14133555

**Published:** 2021-06-25

**Authors:** Aleksander I. Komarov, Lesław Kyzioł, Dmitry V. Orda, Donata O. Iskandarova, Igor A. Sosnovskiy, Artem A. Kurilyonok, Daria Żuk

**Affiliations:** 1The State Scientific Institution “Joint Institute of Mechanical Engineering, National Academy of Sciences of Belarus”, Akademicheskaya Str., 12, 220072 Minsk, Belarus; al_kom@tut.by (A.I.K.); dmitry_orda@mail.ru (D.V.O.); donata_i@mail.ru (D.O.I.); sos3@tut.by (I.A.S.); akto13@mail.ru (A.A.K.); 2Faculty of Marine Engineering, Gdynia Maritime University, Morska St. 81/87, 81-225 Gdynia, Poland; d.zuk@wm.umg.edu.pl

**Keywords:** AlSi12 alloy, centrifugal induction surfacing, coating, lead sublayer, plain bearing, structure, friction coefficient, wear resistance

## Abstract

This paper investigates the structure and mechanical characteristics of a coating based on an AlSi12 alloy, obtained by centrifugal induction surfacing as an alternative to a bronze sliding bearing. To provide for the adhesion of an aluminum layer to the inner surface of a steel bearing housing, a sublayer of low-melting metals was formed, while the formation of the main layer and the sublayer was done in a single processing cycle. The low-melting metals had higher density, which ensured that the sublayer was created at the interface with the steel bearing housing under the action of centrifugal forces. It is shown that the low-melting sublayer forms a strong bond both with the aluminum alloy and with the steel base. Lead and tin are used as low-melting additives. It has been established that lead or tin used in a sublayer are indirectly involved in the structural formation of boundary layers of steel and aluminum claddings, acting as a medium for diffuse mass transfer. Thus, lead is not included in the composition of the main coating and does not change the chemical composition of the aluminum layer. After the addition of tin, the aluminum develops a dendritic structure, with tin captured in the interdendritic space. In this case, the deposited layer is saturated with iron with the formation of intermetallic (Fe, Al, Si) compounds, both at the interface and in the coating volume. This paper offers an explanation of the mechanism through which Pb and Sn act on the structure formation of the coating, and on the boundary layer of the steel bearing housing. Tribological tests have shown that the resulting materials are a promising option for plain bearings and highly competitive with the CuSn10P bronze.

## 1. Introduction

Bronzes belong to a class of materials that have been historically used for the manufacture of plain bearings. However, though they feature good tribological performance, load capacity, and strength, bronzes (particularly those containing tin) are costly.

In addition, bronzes are relatively heavy, which is unwelcome in modern mechanical engineering.

To save bronze, a bimetallic bearing design is usually used, implemented by various technological methods. The cold spray deposition (CSD) method is well-covered in publications. According to the available data, the main disadvantage of this method is the increased porosity of the coatings. Thus, Xueping et al. [1] present their results, according to which the Cu–Sn (6–8%) bronze coating had a porosity of 4.7%. After vacuum heat treatment at 600 °C for 3 h, the porosity decreased to 1.4–2.4%, but the tribological properties deteriorated. An increase in the heat treatment temperature of a tin bronze coating, according to Wen-Ya Li et al. [2], does not eliminate these disadvantages; on the contrary, it worsens its properties due to the active diffusion of the substrate material (Fe and P) with the formation of Fe3P compounds. A decrease in porosity and an increase in the tribomechanical properties of bronze coatings obtained by the CSD method can be achieved by adding dispersed solid particles. Therefore, Xueping et al. [3] showed that the reinforcement of Cu–Sn coatings (8%) with TiN and quasicrystal AlFeCuB particles in an amount of 8.5–9 vol.% provides low porosity (<2%) and high microhardness (230–240 HV0.2) of the coating. In addition, the coefficient of friction and the wear rate are reduced. Similar results were obtained by Wenyuan et al. [4] when reinforcing CuSn5 alloy coatings with Al_2_O_3_ particles. The addition of 1.2–1.8 wt.% Al_2_O_3_ to the coating reduces its porosity to 0.4%, increases the hardness by 20–40 HBW, and improves the tribotechnical properties. It should be noted that the cold spraying method is not suitable for the manufacture of bimetallic plain bearings of small diameters, which is limited by the size of the technological equipment. In addition, according to Wenyuan et al. [4], the efficiency of powder material usage varies in the range of 18–22%, and, according to Xueping et al. [3], no more than 4.5 vol.% of reinforcing particles contained in the charge are embedded in the coating.

One of the widely used methods for the manufacture of bimetallic bushings is the technology of coating surfacing by centrifugal casting. Pavan [5] shows that when pouring an aluminum bronze melt into the inner cavity of a steel billet rotating around its own axis, a uniform distribution of the melt around the perimeter is achieved, and the resulting coating has satisfactory tribotechnical properties. Gafo & Sosnovskii [6,7] provide data on the process modeling of applying bimetallic coatings based on powder materials by centrifugal surfacing technology combined with induction heating. According to these data, the use of induction heating by high-frequency currents ensures uniform heating of the sleeve, while the temperature difference on the inner and outer sides of the sleeve does not exceed 11 °C. Sosnovskii [8] provides data on the modification of a CuSn (10%) P (1%) coating on a steel sleeve with ultrafine boehmite particles, according to which the alloy reinforcement ensures an increase in both the tribological properties and the coating hardness.

However, along with high tribomechanical properties, bronze is expensive. In this context, aluminum alloys can be considered an alternative due to their relatively high specific strength, excellent corrosion resistance, low cost, and advanced technology, and, indeed, they are widely used in the mechanical engineering. Today, we have a wide range of aluminum-based low-friction materials, the high level of tribotechnical properties of which is achieved with the addition of tin, lead, silicon, etc. Thus, Sachek et al. [9] showed that the Al-Si (5%)–Cu (4%)–Sn (6%)–Pb (2%) alloy has tribotechnical properties close to BrOTsT 4-4-17 bronze. At the same time, additional alloying of the alloy with iron (Fe ~1%) provides an increase in the tribotechnical properties up to the level of the antifriction alloy AO3-7.

The main problem faced while designing a bimetallic bearing with an aluminum layer is how to ensure adhesion at the steel-aluminum alloy interface. The low adhesion between aluminum and steel is due to the difference in their thermophysical properties [10], the presence of oxide films, and the occurrence of diffusion processes at the joint interface. One way to solve the problem with adhesion is to form a sublayer between the aluminum coat and the steel base. At the same time, the known method of manufacturing a bimetallic strip containing a sublayer of commercially pure aluminum or zinc coating, applying an anti-friction aluminum coating on a copper base, does not allow obtaining seamless products.

One of the ways to ensure the adhesion of the aluminum layer to the steel base is its aluminization. Thus, according to the results of An et al. [11], the adhesion strength of the Al–Pb alloy with the steel base is increased with the formation of an aluminized layer derived from pure aluminum. In this case, the best effect was achieved when the thickness of the aluminized layer was 73 μm. A high level of adhesion strength between 2024B aluminum and the AZ31 alloy was achieved by Peng et al. [12] by forming a sublayer of Al, Ni and Zn compositions. Similar data are reported in Aceves et al. [13] on the use of a sublayer of Cu, Ti or Ta to directly explosive welding joining aluminum 6061 to 304 stainless steel, here with interlayers that are used to prevent interaction of the aluminum and stainless steel, thus minimizing the brittle intermetallic phase formation. The best results were obtained using Ta, which does not form intermetallic compounds at the phase interface and is highly plastic.

However, the sublayer can be created from materials that do not form intermetallic compounds with aluminum and are denser. In this case, using centrifugal induction surfacing technology, it is possible to obtain an aluminum antifriction layer with a sublayer of a heavier metal or alloy within a single process. In centrifugal induction surfacing, in contrast to the known methods, all components of the composite coating, including the metal of the sublayer and fluxes, are preliminarily laid inside the sleeve, and then the technological parameters are controlled by the power of the HFC inductor and the rotation speed of the sleeve. This method was proposed at the Joint Institute of Mechanical Engineering of the National Academy of Sciences of Belarus by the authors of this article. The material of the sublayer can be lead, tin, zinc, or their alloys, as mentioned by Belotserkovskiy et al. [14]. According to Komarov et al. [15], the use of tin alloys, for example, B83C babbitt or SnPb39 solder, as a sublayer material provides not only the adhesion of the AlSi12 alloy coating to the steel base, but also improves the tribological properties of the coating. In this design, when lead is used, the sublayer material is isolated and not exposed to abrasion, which significantly reduces the environmental footprint of the product—Belotserkovskiy et al. [16].

The aim of this work is to study the structure and tribomechanical properties of the AlSi12 alloy coating with the low-melting sublayer applied on the steel sleeve by centrifugal induction surfacing.

## 2. Materials and Methods

The AlSi12 alloy (wire ER404 AWS A5.10) was used as the material for the coating due to its good casting behavior and mechanical properties, as well as for its satisfactory corrosion resistance at temperatures up to 200 °C. The composition of the alloy is shown in Table 1.

A high percentage of silicon (10–13 wt.%) in the alloy provides good fluidity and casting properties (low shrinkage and practically no cracks), making it possible to lower the casting temperature and increase resistance to heat and wear. In this case, castings from the AlSi12 alloy have low porosity.

To ensure the adhesion of the silumin (AlSi12) cladding to the steel base, a low-melting metal was added in the composition of the charge, which enabled the formation of a sublayer. The quantity of the materials in the charge was measured out to obtain a 3–4 mm thick coat of silumin and a 0.5–1 mm thick sublayer of the low-melting metal.

Lead recovered from waste batteries and tin from Metal Powder Production Plant LLC (Ryazan, Russia) were selected as the materials for the low-melting sublayer. The composition of the mixture of metals is shown in Table 2. Lead and tin have higher densities (11.3 g/cm^3^ and 7.3 g/cm^3^, respectively), and low melting points (327 °C and 232 °C), compared to the AlSi12 surfacing aluminum alloy (2.7 g/cm^3^, and 578 °C) [17].

The coating on the inner surface of a cylindrical billet made of steel (EN 10250–2) [18] was formed by the centrifugal molding of a bulk mixture consisting of the AlSi12 alloy, the low-melting metal supposed to form the sublayer, and a coating flux (see Table 3), in an amount of 5 wt.% AlSi12 alloy.

The heating was done with RF currents to a temperature of 780–800 °C at a rotation speed of 1500 rpm. After reaching the specified process parameters, the isothermal exposure was applied for 5 min, followed by the accelerated cooling to a temperature of 150–200 °C. The surfacing scheme is shown in Figure 1.

The centrifugal induction surfacing was carried out using equipment including an RF current generator (LPZ-2–67M, 60 kW, 66 kHz), a frame inductor, a program controller (TRM 151 from Owen, Moscow, Russia), and a centrifugal adjustable-speed induction surfacing station (designed by Joint Institute of Mechanical Engineering of the NAS of Belarus, Minsk, Belarus). The temperature was measured during the heating with a pyrometer (TemPro-2200, ADA Instruments, Moscow, Russia). The heating time was controlled by an electronic stopwatch.

The structural phase state of the obtained samples and friction surfaces was studied using an MIM-8 metallographic microscope and a VegaTeskan scanning electron microscope (TESCAN, Brno-Kohoutovice, Czech Republic). The microhardness of the combined aluminum coating with a low-melting sublayer was determined on a PMT-3 device (LOMO, City, Russia) under loads of 0.196 and 0.49 N.

The tests were carried out by extruding the aluminum cladding from the steel sleeve (the test scheme is shown in Figure 2a). The loading rate was 5 mm/min. The force of separation (or peel force, P_o_) of the AlSi12 alloy coating from the steel base was recorded upon indications of the test machine. The adhesion strength (σ_a_) was computed from the known value of P_o_ and the parameters of the initial billet (D—inner diameter and h—height): σ_a_ = P_o_/F = P_o_/(πDh), where F is the contact area of the coating with the base, which was defined as a product of the circumference of the steel billet (πD) and the height (h) of the test sleeve.

The peel force (P_o_) was determined from the tensile strength diagram of the AlSi12 alloy coating sample with the low-melting steel-facing sublayer, which corresponds to the tensile strength at the time of the peeling the coating off the steel base.

Tribological tests of the samples were carried out on a multifunction tribometer (MFT-5000 from Rtec Instruments, Oakland, CA, USA) according to the scheme of reciprocating movement of the sample 2 (see Figure 2b) against the fixed steel counterbody 1. The counterbody is the end surface of the cylinder made of 100Cr6 steel (hardness 60–62 HRC) with a diameter of 3 mm. The tests were carried out under loads of 40, 80 and 120 N. The amplitude of the movement was 5 mm and the frequency was 10 Hz. The duration of the tests in the loaded state was 2 h. Based on the test results, the coefficient of friction (f) and volume (Iv) of wear intensity (Iv = Δv/N·L, where Δv is volume wear, L is the total friction path, N is load) were determined.

Volumetric losses (Δv) of each sample were determined as Δv = Δm·ρ, where ρ is the density of the material area on which the tribotechnical tests were carried out, Δm is the weight loss when weighing on an analytical balance (VLR-200, Mettler Toledo, Moscow, Russia). For samples with a lead sublayer, in the calculations, we used the density values of the AlSi12 alloy, the structure in the friction zone which corresponds to the reference data. For samples of coatings with the tin addition, the density calculation of the surface structure was carried out on the basis of the X-ray analysis data, proceeding from the ratio of the structural components, of which the densities are known. The total linear wear of the friction pair was determined by the change in the position of the load sensor.

## 3. Results and Discussion

### 3.1. The Coating Structure with the Lead Sublayer

A metallographic analysis of the AlSi12 alloy coating on a sample with a lead sublayer revealed that lead does not affect the coating structure and its formation, which is due to the workpiece cooling rate. The deposited coating has a eutectic structure (see Figure 3a). As we approach the interface with lead, the particle size of eutectic silicon increases, which is associated with a longer exposure to the elevated temperature in this area (see Figure 3b). Iron-containing inclusions of a needle-shaped form are evenly distributed over the alloy structure, but their size slightly increases as they approach the steel base (see Figure 3b).

As the results of the metallographic analysis and SEM have showed, the lead sublayer forms developed interfaces both with the aluminum alloy and with the steel sleeve (see Figure 3b and Figure 4). In this case, there are no inclusions in the form of oxide films, which may be due to the reducing effect of carbon introduced into the mixture at the lead preparation step (see Figure 4). The remaining carbon is released in the lead layer as uniformly distributed spherical inclusions of 2–5 µm (see Figure 4 and Figure 5).

An interesting feature of the material structure formation of the system under consideration, which contains the lead sublayer, is a product of the aluminized diffusion layer set on the steel sleeve surface. Its structure is represented by elongated grains (up to 250 μm in length) directed along the normal to the surface (see Figure 3d and Figure 6). Thus, the steel (aluminized layer) is a lead interface that has a developed surface formed as a result of the dissolution of iron from the aluminized layer (see Figure 6). However, according to the results, when tin is added to the composition of the sublayer material, the aluminized layer is not formed (see Figure 7). Moreover, the structure of the deposited aluminum alloy differs significantly from that considered previously. In this case, there is no distinct tin sublayer, which is associated with its lower density (7.31 g/cm^3^ for tin and 11.34 g/cm^3^ for lead). In addition, as the chemical analysis shows, there is an active saturation of the AlSi12 melt with iron. Where the iron content in the aluminum coating with the Pb sublayer does not exceed 0.5 wt.%, its content in the surface with the tin sublayer reaches 12 wt.%. In this case, iron forms inclusions of the Fe–Al–Si system, which form an intermediate layer at the interface with steel, and also stand out as polyhedra in the deposited layer structure (see Figure 7).

When different metals are used to form the sublayer, this structure formation feature of the interface zone can be explained as follows: when lead is used for the sublayer, according to the dual state diagram of Al-Pb in Lyakishev [19], at a temperature of 800 °C, the melt stratifies into two liquids, which are Al (Pb) (Pb melt in Al, about 4 at.% Pb) and Pb (Al) (Al melt in Pb, about 0.5 at.% Al). Since the coating surfacing is carried out under the action of centrifugal forces, the denser melt, Pb (Al), is pushed to the boundary with the steel sleeve. As follows from Lyakishev [20], iron in the Pb-Fe system has a low solubility in lead, which is 0.01 at.% at a temperature of 800 °C. Therefore, the lead melt serves as a kind of a barrier and prevents the transition of iron into aluminum, but at the same time, aluminum dissolved in lead diffuses into the surface of the steel base, forming an aluminized layer with a structure characteristic of liquid aluminization.

In contrast to the case considered, when tin is used for the sublayer and the mix is fully melted, a single liquid of the Al–Sn–Si system is formed. Since, according to the state diagrams of Sn–Fe [20] and Al–Fe [19], the solubility of iron in tin and aluminum is 3.2 at.% and 5 at.%, respectively, at a temperature of 800 °C, the melt is actively saturated with this element. The intensive dissolution of iron competes with the diffusion of aluminum into the surface of the steel sleeve, which explains the absence of the aluminized layer in this case. When cooling the sleeve, and during the action of centrifugal force, aluminum and tin, which have different melting points, are separated, forming a dendritic structure of a solid solution of aluminum with tin in the interdendritic space and inclusions of iron-containing intermetallic compounds (see Figure 6). According to the XRD (see Figure 8) and scanning microscopy data, these compounds are phase (Fe, Al, Si) with different stoichiometric compositions. Silicon is also represented by the α + Si eutectic, as well as its primary crystals.

The results of the X-ray structural analysis also confirm the assumption made about the mechanism of iron diffusion into the aluminum coating and correspond to the data of the scanning microscopy. Thus, the diffraction pattern of the coating with the lead sublayer (see Figure 8a) contains reflections from the iron-containing phase β-Al5FeSi of low intensity, while in the coating with tin, the intensity of reflections from iron-containing inclusions is higher (see Figure 8b).

X-ray diffraction analysis also revealed the presence of a small amount of lead in the structure of the coating with the lead sublayer (see Figure 8a), which could not be detected at the metallographic study stage. As shown by the results of the scanning microscopy, lead is represented by globular inclusions ranging in size from 0.5 to 5 µm, uniformly distributed in the structure of the aluminum alloy (see Figure 9, indicated by arrows). The presence of such inclusions in addition to the formed lead sublayer can be explained by crystallization processes in the Al–Pb system. According to the phase diagram of the Al–Pb system [19], the final phase of crystallization at a temperature of 659 °C is characterized by the formation of a melt of eutectic composition containing 0.17–0.19 at.% (≈1.5–2 wt.%) Pb, from which globular lead inclusions are distinguished.

### 3.2. The Adhesion Strength and Microhardness

Our analysis of the peeling-off zone structure of the samples of the combined coating with the lead sublayer showed that the separation of the coating occurs mainly along the thickness of the lead sublayer, while the value of the adhesion strength (σa) of the coating to the steel base is 9–10 MPa. Considering that the lead tensile strength is 11–13 MPa, we can say that the value of adhesion is approximately equal to the tensile strength of the lead sublayer. Since the steel is not wetted by the lead melt under normal conditions, the sufficiently high adhesion of the lead sublayer to the steel sleeve can be explained by the formation of an aluminized layer during surfacing. The presence of an aluminized layer as a condition for ensuring high adhesion during the deposition of the Al–Pb alloy on a steel base was also shown in the works of An et al. [11,21]. In this case, the adhesion strength increased to a certain value of the aluminized layer thickness, and then decreased, which is associated with its considerable fragility.

When tin is used as an additive, the average adhesion strength is 36 MPa, which significantly exceeds the ultimate strength of tin (Rm = 19–21 MPa). This can be explained by the process nature of structure formation of the deposited aluminum layer with the addition of tin. As shown earlier, in this case there is no pronounced tin sublayer, and the separation occurs at the interface of the steel sleeve with a coating, the strength of which is likely to exceed the strength of tin.

Our measurement of the microhardness on the span from the surface of the AlSi12 alloy coating to the steel base showed that its values vary from 450 MPa (for eutectic at the interface with the lead sublayer) to 650 MPa (for eutectic on the surface). When approaching the boundary with the lead sublayer, the value of hardness does not change. The lead sublayer has a microhardness of 70–85 MPa throughout its thickness. The aluminized layer is characterized by a high hardness at a level of 7.7 GPa; however, at the boundary with the lead layer, in the dissolution zone, it decreases to 5–5.5 GPa, which may be due to the presence of lead in the structure. As one approaches the steel base, the hardness of the diffusion layer decreases slightly and has values of 6.8–7.0 GPa. The steel base is characterized by a uniform value of microhardness within a range of 1000–1050 MPa.

When tin is used as the additive, the microhardness of the eutectic structure is comparable to the values measured for the samples with the lead sublayer. However, the structure of the coating contains iron-containing inclusions, the microhardness of which varies within a range from 5.5 to 7 GPa, while the equiaxed particles have a higher hardness due to the increased iron content in their composition. It can be expected that such a structure with inclusions of hard particles in a softer, tin-containing matrix will improve the tribotechnical properties of the deposited material.

Analysis of the microhardness of the coating with the tin addition showed that the hardness of the areas of the eutectic structure is 362–417 MPa, while the areas with the tin inclusion have a lower hardness of 175–247 MPa. The needle-shaped iron-containing inclusions have a hardness of 6.5–7.3 GPa, while such inclusions at the boundary with the steel sleeve have an increased hardness (up to 10.5–14.0 GPa), which is probably associated with an increased iron content. Based on the data obtained, it can be concluded that when tin is used as a low-melting additive, the formation of a structure is observed according to the Charpy principle (there are solid inclusions in the soft matrix), which provides increased antifriction properties.

### 3.3. The Tribological Properties

According to the test results, the friction coefficient of the AlSi12 alloy coating with the lead sublayer varies within a range from 0.019 to 0.034 under different loads (see Figure 10). The highest values of the friction coefficient for this material are recorded under low loads, which may be associated with slow run-in processes. The load increase is initially accompanied by a friction coefficient decrease; however, at 120 N, an increase in the friction coefficient is observed in connection with a growing role of adhesive interaction processes and the destruction of brittle silicon inclusions.

When tin is used as the additive, the values of the friction coefficient are approximately 2 times lower than in the previous case (see Figure 10b), which is probably due to the favorable effect of tin on the surface structure formation during friction (see Figure 10). In this case, a layer is formed on the surface that protects the materials of the pair from wear and provides a low friction coefficient. Further, in contrast to the previous case, traces of chipping of the solid inclusions are observed to a much lesser extent. The formation of such a structure can be explained by the tin presence on the surface of the iron-containing phases, which contributes to their retention in the aluminum matrix, while acting as a solid lubricant.

With the load increase, an insignificant grow in the friction coefficient is observed, which is probably associated with the destruction and partial spalling of brittle phase fragments (Fe, Al, Si) (see Figure 11). Despite this, the presence in the friction zone of hard iron-containing inclusions and soft particles of tin or lead alloy provides an increase in the wear resistance of the deposited coating. Thus, the wear rate of the sample with the lead sublayer was 2.18 × 10^−6^ mm^3^/m·N, while, for the coating sample with the addition of tin, this value was 6.53 × 10^−7^ mm^3^/m·N, which is comparable to the data for the bronze alloy CuSn10P (Polat et al.) [22].

## 4. Conclusions

It is shown that lead, like tin, can be used as a material for a low-melting sublayer in the manufacture of a combined sleeve with a functional layer from the AlSi12 aluminum alloy.

Lead, while adhering to both the aluminum coating and the steel substrate, acts as a medium for transferring aluminum to the steel surface, forming an aluminized layer with a high microhardness of 7.0–7.7 GPa. The aluminized layer plays an important role in ensuring the adhesion of the lead sublayer to the steel sleeve, with the average adhesion strength being 9–10 MPa, which is comparable to the tensile strength of lead. The adhesion of the deposited coating with the addition of tin has even higher values (36 MPa) and is due to the absence of an obvious tin sublayer.

The structural analysis of the AlSi12 alloy coating showed that lead does not affect the processes of structure formation, which are caused by the cooling rate of the workpiece. The resulting structure provides a low coefficient of friction, from 0.019 to 0.034, and a wear rate of max. 6.53 × 10^−7^ mm^3^/m·N.

When tin is used, there is a tendency to the saturation of the melt with iron, as a result of which solid inclusions of the Fe–Si–Al system are present in the alloy structure after crystallization. This structure provides better tribotechnical surfacing characteristics comparable to CuSn10P bronze coatings.

## Figures and Tables

**Figure 1 materials-14-03555-f001:**
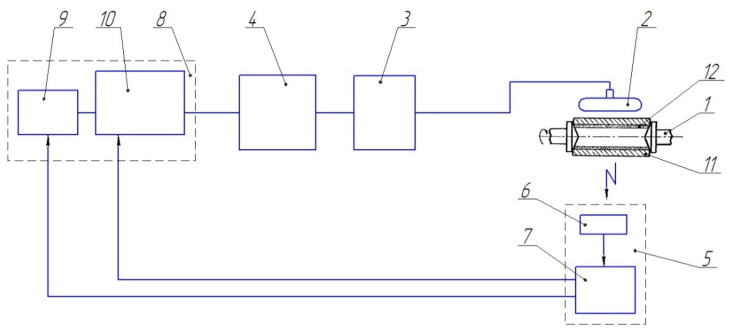
Scheme of induction surfacing of samples: 1—clamping centers; 2—a frame arc-shaped inductor; 3—heating source; 4—power regulator; 5—temperature measurement system; 6—primary pyrometric converter; 7—secondary measuring transducer; 8—temperature controller; 9—digital converter of the programmer; 10—matching unit; 11—blank; 12—weldable powder charge.

**Figure 2 materials-14-03555-f002:**
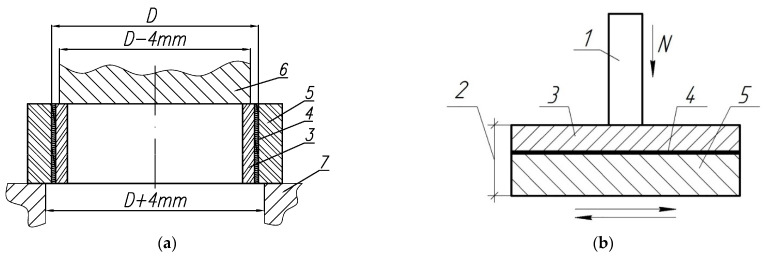
The schemes of adhesive (**a**) and tribological (**b**) testing of the samples: N—load; 1—counterbody; 2—sample; 3—AlSi12 alloy coating; 4—transition layer; 5—steel base; 6—punch; 7—matrix form.

**Figure 3 materials-14-03555-f003:**
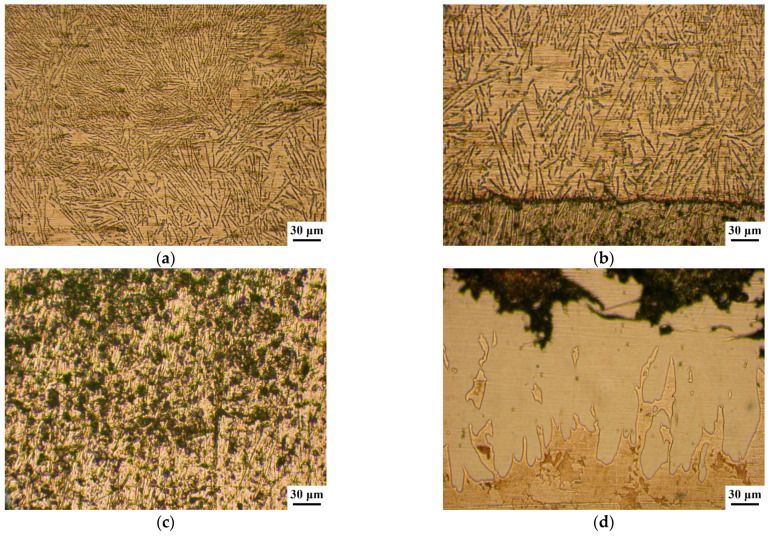
Structure of a combined coating of the AlSi12 alloy with a lead sublayer: (**a**) the structure of the middle part of the coating; (**b**) the coating structure at the border with the lead sublayer; (**c**) the sublayer structure; (**d**) the structure of the diffusion layer on the steel base.

**Figure 4 materials-14-03555-f004:**
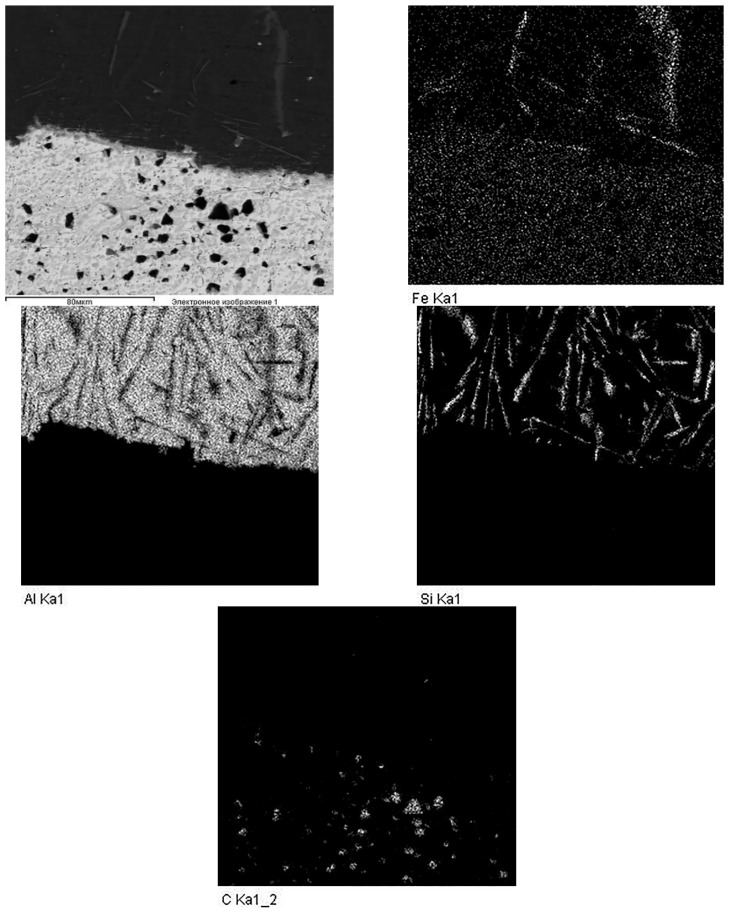
Elements distributed on maps in the deposited coating with the lead sublayer.

**Figure 5 materials-14-03555-f005:**
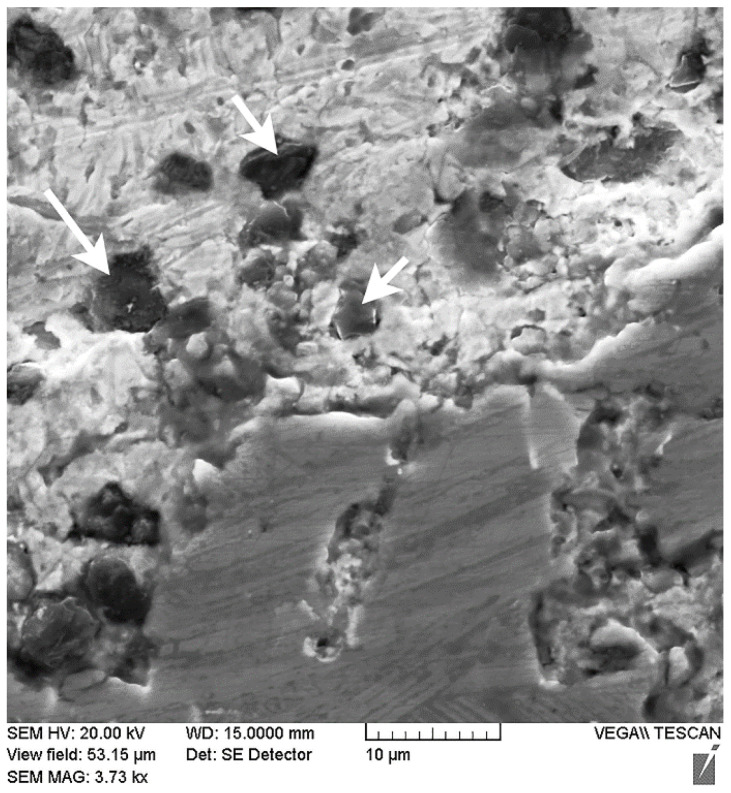
The structure of the lead sublayer with carbon inclusions (indicated by arrows).

**Figure 6 materials-14-03555-f006:**
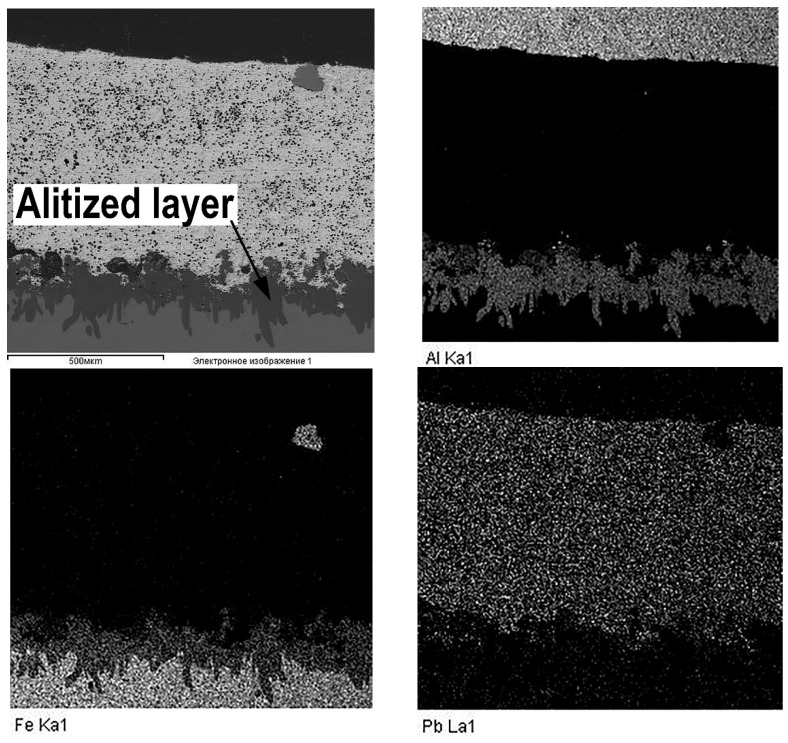
The alitized layer at the boundary of the lead sublayer and the steel base.

**Figure 7 materials-14-03555-f007:**
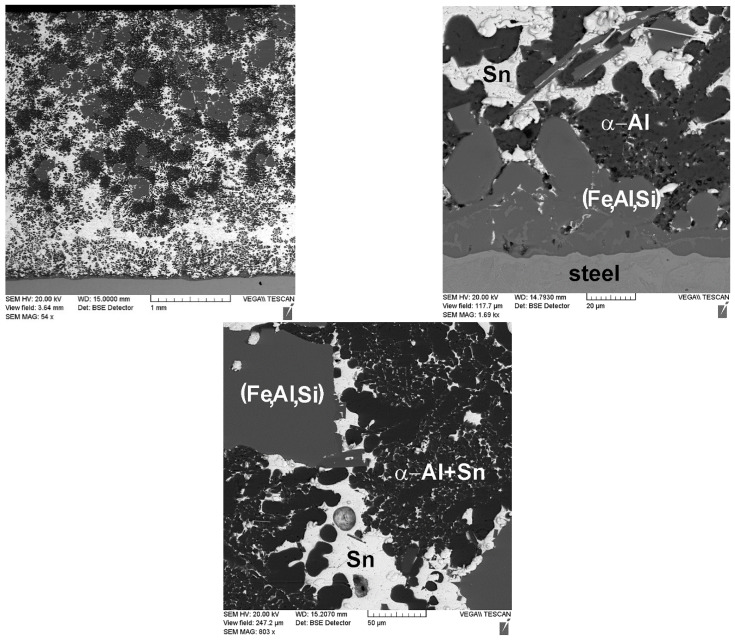
The microstructure of the deposited layer from the AlSi12 alloy with the tin addition.

**Figure 8 materials-14-03555-f008:**
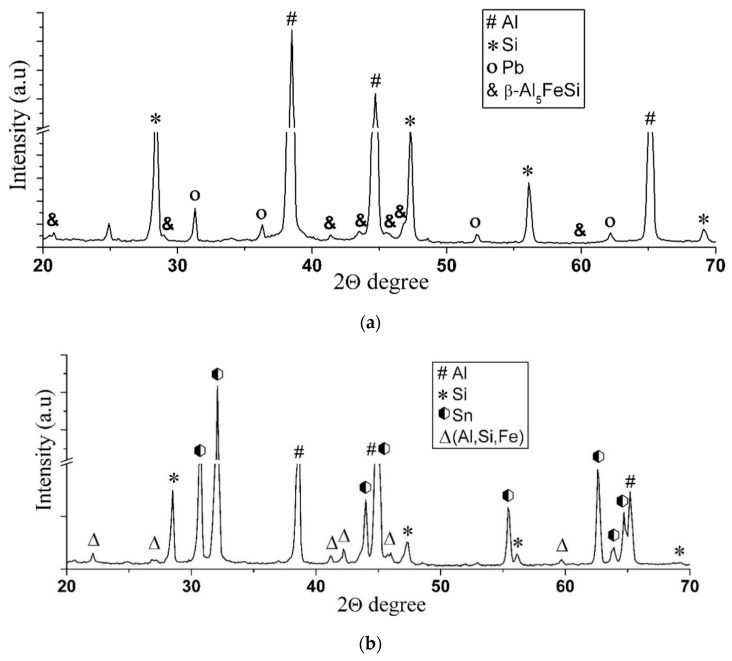
The diffraction pattern of a coating made of the AlSi12 alloy with a lead sublayer (**a**) and with the addition of tin (**b**).

**Figure 9 materials-14-03555-f009:**
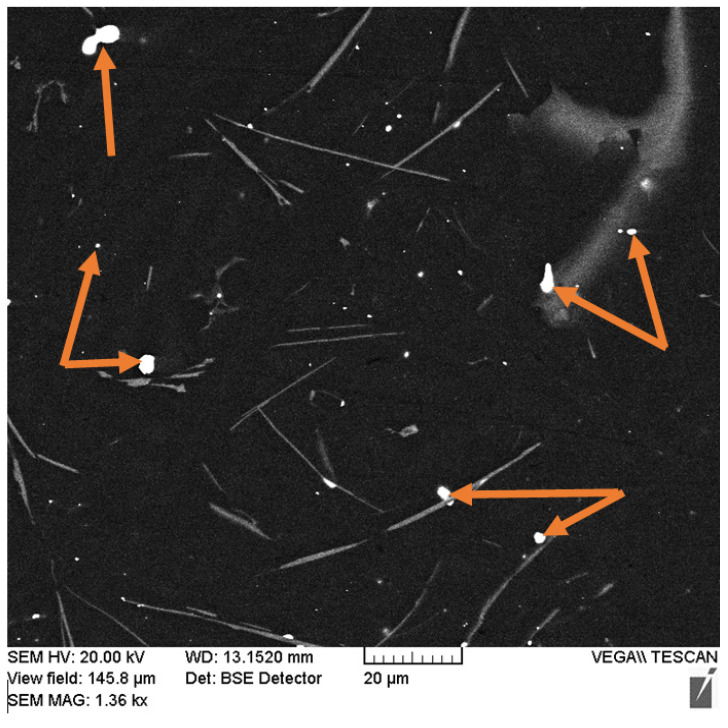
The microstructure of the middle part of the coating from the AlSi12 alloy with the lead sublayer.

**Figure 10 materials-14-03555-f010:**
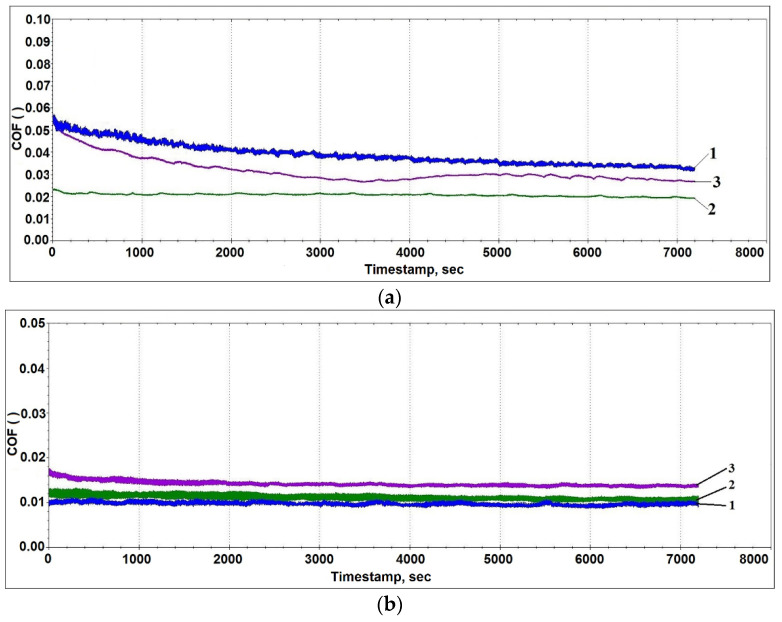
The change in the friction coefficient during the testing of the combined coating made of the AlSi12 alloy with the lead sublayer (**a**) and with the addition of tin (**b**) under different loads: 1—40 N; 2—80 N; 3—120 N.

**Figure 11 materials-14-03555-f011:**
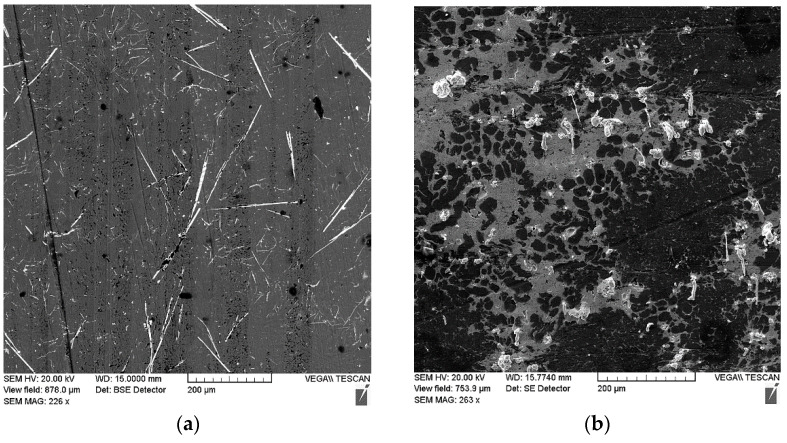
The structure of friction surfaces after tests performed under a load of 120 N on the samples of the combined coating made from the AlSi12 alloy with the lead sublayer (**a**) and with the addition of tin (**b**).

**Table 1 materials-14-03555-t001:** AlSi12 alloy composition (wt.%).

Si	Fe	Cu	Mn	Ti	Mg	Zn	Al
12.1	0.18	0.05–0.15	0.2–0.4	up to 0.01	0.08	up to 0.004	rem.

**Table 2 materials-14-03555-t002:** The mix of metals for the sublayer (wt.%).

Metal	Sb	Cu	Fe	As	Zn	Sn	Pb
Pb	The total fraction of impurities is not more than 12%						remain
Sn	0.011	0.002	0.004	0.003	<0.001	rem.	0.01

**Table 3 materials-14-03555-t003:** The composition of the coating flux (wt.%).

KCl	NaCl	NaF	Na_3_AlF_6_
10	50	30	10

## Data Availability

Data sharing is not applicable to this article.

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
