# Peer review of "Creation of AlSi12 Alloy Coating by Centrifugal Induction Surfacing with the Addition of Low-Melting Metals"

_materials, 2021, doi:10.3390/ma14133555_

Round 1
Reviewer 1 Report
The authors present data on the structure and performance of aluminum bearing surfaces bonded to steel cylinders using two intermediate materials (lead and tin) and centrifugal induction melting. Results show distinctly different microstructures for the two materials as well as very different tribological performance. Overall, the paper is very interesting. Some revisions are required to make it more clear.
Statement on lines 68/69 is unclear - earlier it was stated that the coatings had 1.2-1.8 wt%, but now it appears that the original powers for the coating process had 50 volume%? I am not sure if the sentence is supposed to say 9-50 volume percent or 9% of 50% (i.e. 4.5%). Please review and revise.
The testing description of the peel strength (lines 176-184) is somewhat unclear. How are the two components (the steel sleeve and the aluminum inner coating) held during the extrusion process? Given the fact that the structure of the coating and intermediate layer varied significantly between the tin and lead additions, was any thought given to whether the geometry of the fixturing for the extrusion (for example, the application of clamping force relative to the intermediate layer) could have affected the measured separation force?
Is the volume loss calculation (lines 200-203) accurate for both compositions? Part of the explanation for the difference in tribological properties appears to be a difference in inclusions in the outer layer of the coating; however, the volume loss is calculation assuming the material lost is only the AlSi alloy.
It would be helpful to provide a clearer explanation (or even a schematic diagram) of the locations corresponding to where images 3-6 were taken. After reading the descriptions multiple times and looking over the images repeatedly, I THINK image 4 is intended to show the Al/Pb interface details and image 6 is intended to show the Pb/Fe interface details but I am not certain. More clarification is needed.
What is the definition of the 'MKM' label in the legend for Figure 3? Figure 5, for example, is labeled in micrometers. A consistent labeling scheme should be used, if possible.
Lines 307-312 - what is the uncertainty on the measurements for the strength of adhesion? Do the error bars of the two strengths actually overlap (justifying the statement they are equal) or is the adhesive strength statistically lower than the tensile strength?
With regards to the COF trend in Figure 10a, did examination of the micrographs of samples tested at 40N/80N/120N support the explanation presented in lines 351-357?
Were the values of wear rates provided (e.g. lines 379-380) measured at a certain applied load? The text appears to imply the wear rate increased with load, but it is not clear what load corresponds to the stated values.
Was any investigation made regarding the presence of elements from the flux in the final coating structure?
Were any longer duration tests done to compare fatigue/spalling characteristics of the two coatings? That is likely beyond the scope of the present paper.
Grammar is awkward in some places - please review.
Author Response
The authors would like to thank the reviewer for the valuable comments. These comments improve the quality of the article.
Yellow color highlighting in the text marks corrections and responses to reviewer.

Reviewer 2 Report
The study presented the characterization results of the structure and mechanical properties for a coating based on AlSi12 alloy as prepared by the centrifugal induction surfacing. Low-melting metals were used in the sublayer formation. Tribological tests showed promising results for the fabricated coating on the steel bearing. Several critical issues need to be addressed before publication.
- The novelty and contribution of the selected coating materials and approaches need to be more clearly indicated in comparison with previous studies. For example, why the centrifugal induction surfacing? Has a similar material system been applied before?
- More details of the recovered lead can be provided. Will it potentially affect the obtained mechanical properties?
- Are there any potential issues of using the low melting metals?
- The formats and English writing can be improved.
Author Response
The authors would like to thank the reviewer for the valuable comments. These comments improve the quality of the article.

Round 2
Reviewer 2 Report
The manuscript is ready for publication.